# Psycholinguistic Diagnosis of Language Models' Commonsense Reasoning

**Yan Cong**

yancong222@gmail.com

## Abstract

Neural language models have attracted a lot of attention in the past few years. More and more researchers are getting intrigued by how language models encode commonsense, specifically what kind of commonsense they understand, and why they do. This paper analyzed neural language models' understanding of commonsense pragmatics (i.e., implied meanings) through human behavioral and neurophysiological data. These psycholinguistic tests are designed to draw conclusions based on predictive responses in context, making them very well suited to test word-prediction models such as BERT in natural settings. They can provide the appropriate prompts and tasks to answer questions about linguistic mechanisms underlying predictive responses. This paper adopted psycholinguistic datasets to probe language models' commonsense reasoning. Findings suggest that GPT-3's performance was mostly at chance in the psycholinguistic tasks. We also showed that DistillBERT had some understanding of the (implied) intent that's shared among most people. Such intent is implicitly reflected in the usage of conversational implicatures and presuppositions. Whether or not fine-tuning improved its performance to human-level depends on the type of commonsense reasoning.

## 1 Introduction

In this paper, we focus on Language Models' (LMs) performance in commonsense reasoning tasks. Different from language semantics concerning logical relations between isolated sentence meanings, we take pragmatics to be sentences' relations relying on conversational participants' commonsense, such as the basic level *intent* that is commonly shared among most people. Humans reason about what their interlocutor could have said but chose not to, thereby drawing various inferences. The way humans put linguistic meanings to use depends on social interaction and commonsense assumption. What about machines whose pre-trainings do not involve social interaction? To what extent do they still have this pragmatic knowledge? How do they cooperate without any forms of learning in Grice pragmatics (Grice, 1975)? This paper attempts to answer these questions by examining transformer LMs' performance in commonsense reasoning.

We focus on two commonsense pragmatics phenomena: (i) Presupposition (henceforth Presp), for example, by using determiner *the* in the utterance "*the* teacher spoke to me" most people typically presuppose the existence of such a teacher in the context; (ii) Scalar Implicature (henceforth SI), for example, by using quantifier *some* in "I ate *some* of the cookies", most people generally imply "not all". We provided linguistic perspectives about how humans compute and evaluate commonsense pragmatics. We then assessed the extent to which LMs can understand the meanings pragmatically enriched by human speakers. Moreover, we fine-tuned LMs with pragmatic inference datasets. Evaluation comparisons are reported and discussed. We make all code and test data available for additional testing[1].

## 2 Related work

LMs' knowledge about syntax and semantics is relatively well studied (Warstadt et al., 2020; Tenney et al., 2019; Devlin et al., 2019). Considerably fewer studies have been done on speaker's intent: the implied meaning that's commonly shared among most people's intention. This is called Conversational Implicature in pragmatics literature (Grice, 1975). Implicature phenomena like quantifiers *some* and *many* are tested in recent studies (Schuster et al., 2020; Jeretic et al., 2020). The diagnostics in these studies are controlled. Most of them incorporate offline human responses to words in context such as acceptability judgment surveys.

Relatively few studies include online human response in the assessment (Ettinger, 2020). On-

---

[1] https://github.com/yancong222/
Pragamtics-Commonsense-LMs

line measurement uses neurolinguistic equipment electroencephalogram (EEG) and Event-Related-Potentials (ERP) to record brain activity (Luck, 2012). ERP components such as N400 wave is an event-related brain potential measured using EEG. N400 refers to a negativity peaking at about 400 milliseconds after stimulus onset. It has been used to investigate semantic processing. N400 is relevant because it's an online real-time measurement of human brain's response to different language phenomena, and it has been mostly elicited as a result of human processing sentences with semantic anomalies. Online measurement differs from offline judgments survey or cloze test in that online measurement reveals human brain's real-time sensitivity to (linguistic) cues. We examine LMs using human centered datasets that are collected through both offline and online experiments.

How "human-like" the state-of-the-art LMs are (cognitive plausibility) has not comprehensively justified (Wang et al., 2019). Goldstein et al. (2021) provides empirical evidence that the human brain and GPT-2 share fundamental computational principles as they process natural language. In a sense that both are engaged in continuous next-word prediction, and both represent words as a function of the previous context. Against this background, we study LMs' cognitive plausibility through examining their performance in understanding pragmatically enriched meanings, which are *implied* or *presupposed* among most people (i.e. conversational participants) to convey their intentions.

## 3 Experiments

We first designed most of the tests in the form of cloze tasks, so as to test the pre-trained LMs in their most natural setting, without interference from fine-tuning. The main schema we used in this study is called the *minimal pair paradigm*, in which two linguistic items are in contrastive distribution, meaning the two items are identical except one single aspect. The notion of *minimal pair* is widely used in linguistic experiments probing the underlying structures of a linguistic utterance. Typically, one of the two items is pragmatically *odd* according to most people's commonsense knowledge (marked by #), relative to the other utterance in the minimal pair.

The hypothesis and the accuracy calculation pipeline are as follows. If LMs understand commonsense intent, which gets reflected in the usage

| Model | $n_{\text{params}}$ | $n_{\text{layers}}$ |
|---|---|---|
| DistillBERT-base-uncased | 67M | 6 |
| GPT-3/InstructGPT | 175.0B | 96 |

Table 1: (pre-trained LMs) Model cards

of SI and Presp, LMs should endorse more often the pragmatically good sentence than its pragmatically odd counterpart in a minimal pair. To quantify such "endorsement", we calculated the percentage $p$ of cases in which LMs favor the pragmatically good sentence over the pragmatically odd one. The extent to which LMs (dis-)favor an sentence is derived from LMs' tokenized sequence log probability (henceforth logprob). The accuracy mean for each condition (*good* vs. *bad/so-so*) is then calculated per phenomenon (SI and Presp), using the sum of percentage $p$ divided by the number of sentences, grouped by phenomenon. DistillBERT (Sanh et al., 2019) is used, which has only the *encoder* transformer, It's necessary that models are able to use right-hand context for word predictions. We compare DistillBERT with another type of LMs GPT-3 (Brown et al., 2020), which has only the *decoder*. We present model cards in Table (1).

**Study 1: Presupposition** Our first study is built up on Singh et al. (2016). They performed human behavioral acceptance judgment experiments using the presupposition triggers *the*. Participants were asked to drop out when they think the sentence stops making sense. Singh et al. (2016)'s findings show that humans think utterances make less sense relative to the controls when the presupposed information is implausible. We extracted 82 items from Singh et al. (2016) human experiments stimuli, which are already cognitively justified and freely available in their appendix. *Seth went to jail/ # a restaurant on Saturday night. The guard spoke to him there for a while.* presupposes that *there is a unique* guard in the context. Given common-sense world knowledge and the close association of guard and jail, "Seth went to jail" is a more likely and plausible context, thus "a restaurant" is marked with #. Utterance *Kristen went to a restaurant/ # jail in the morning. The waiter served her there quickly.* presupposes *the existence* of a (unique) waiter in the context. "Kristen went to a restaurant" is a better context in a sense that it lays out

a background where there is a waiter. By contrast, jail is rarely associated with waiter, "went to jail" is implausible and is marked with #. It's both the uniqueness of the "waiter" and the relevance of the job to the place "restaurant" that affect the context. Singh et al. (2016) reported that in this stops-making-sense paradigm, human participants were near-ceiling in accepting plausible conditions: at the last region of the sentence, the acceptance rate was 95% in the plausible condition. For implausible *the*, by the end of the sentence, 50% dropped out since it stops making sense and most people cannot accept it.

Built up on Singh et al. (2016) human experiment, we evaluated LMs' sensitivity to Presp. We compared the accuracy mean of each condition, as exemplified in *John went to school on Monday afternoon. The substitute teacher spoke to him there briefly.* versus *John went to a concert on Monday afternoon. The substitute teacher spoke to him there briefly.*. The two utterances differ in only one element "school"/"concert". The former is pragmatically good relative to the latter, given that *the* presupposes a context where *there is* a teacher, and commonsense tells us that "teacher" and "school" are closer than "teacher" and "concert".

GPT-3 is evaluated by the extent to which it favors plausible cases over the implausible ones. Sequential word-by-word logprob is generated and transformed into percent. We take the sum of word level logprob averaged by sentence length to be a proxy to the sentence *naturalness*. Higher percent indicates that GPT-3 evaluates the sentence to be natural. DistillBERT is evaluated through critical word prediction. Noun phrase in the initial sentence is masked and taken as the critical word. (e.g., *school* is masked in "*John went to school. The substitute teacher spoke to him there briefly.*", whereas *concert* is masked in "*John went to a concert. The substitute teacher spoke to him there briefly.*". Given that human data shows preference to the plausible over the implausible, DistillBERT is considered to have succeeded if the critical word is in its top$K$ ($K$=5) tokens for the plausible sentence. It's also considered succeed if the critical word is NOT in BERT's top$K$ for the implausible sentence.

**Study 2: Scalar Implicature** According to Nieuwland et al. (2010), relative clauses can make implicatures unnoticed by most people in sentence processing. Table (2) shows that there is a prag-

matic violation in (a) if conversation participant actively draws pragmatic inference that "some (but not all)" office buildings have desks. However, this violation is left unnoticed in (a) due to the presence of the relative clause. (c) is relatively bad and implausible compared to (d): the violation in (c) is noticed due to the absence of a relative clause. Note that Nieuwland et al. (2010) considered the Communication sub-scale of the Autism-Spectrum Quotient questionnaire (AQ) (Baron-Cohen et al., 1994, 2001; Baron-Cohen, 2008) to be a proxy to be an individual's pragmatic skills. According to Nieuwland et al. (2010), the AQ quantifies pragmatic capabilities on a continuum from autism to typicality.

Nieuwland et al. (2010) reported that only pragmatically skilled participants (i.e., lower autism scores) are sensitive to the pragmatic violation in (c) ($r$=-.53, $p$=0.003). For (a), in which the implicature is left unnoticed, so is the violation. There is thus no significant difference between the pragmatically skilled participants and those who have high autism scores ($r$=-.29, $p$=0.13). Overall pragmatically skilled people are good at generating robust pragmatic inferences that *some* implies *not all*, which gives rise to larger N400 when the utterance is pragmatically bad - N400 is a verified ERP elicited by anomaly stimuli (Luck, 2012).

We extracted 168 items from Nieuwland et al. (2010). Some examples of items from their data are "*Some people have lungs/pets, which require good care*". GPT-3 is used for sequential word prediction. Using sum of token level logprob averaged by sentence length, we examine if there is a difference with and without the SI being noticed. GPT-3 is considered succeed if the plausible sentence mean is higher (hence more favorable) than the soso/unacceptable sentence mean. We use masked language models like DistillBERT for critical word prediction. We masked quantifiers and take *some* as the critical word for (a,b,d). We take *all* as the critical word for (c), because SI is noticed and *all* is commonsense intent. Now that (a,b,c,d) are all not implausible, BERT is marked as succeed if the critical word is in its top$5$ tokens list.

**Sanity check** One may wonder to what extent LM is merely leveraging nouns joint-probability. This motivates us to check whether the test datasets contain enough noun co-occurrence patterns that could make the LMs find a likelihood pattern rather than actually *reason* to conclude which sentence

| Plausibility | Example | Label |
|---|---|---|
| So-so | (a) [Some] office buildings have *desks* that are covered with dust. | SI unnoticed |
| Plausible | (b) [Some] office buildings have *plants* that are covered with dust. | SI unnoticed |
| Implausible | (c) [Some] office buildings have *desks* and can become dusty. | SI noticed |
| Plausible | (d) [Some] office buildings have *plants* and can become dusty. | SI noticed |

Table 2: Datasets and examples used in SI evaluation (Nieuwland et al., 2010)

is more plausible. For instance, the co-occurrence of *office-buildings* and *desks* in the SI *good* pair seems to be more frequently seen than that of *office-buildings* and *plants* in the *bad* pair, since plants are not essential, but desks are. Similarly, for the Presp stimuli, it appears that humans tend to associate *jail* with *guard* more frequently than they do so for *restaurant* and *guard*. To address these confounding factors, we use n-gram to calculate joint-probability (Yin et al., 2016). Results show that 70% of the SI and 50% of the Presp stimuli show higher co-occurrence probability in the 'good' sentence than in the 'bad' sentence[2].

## 4  Fine-tuning DistillBERT with ImpPres

In order to examine how to improve LMs' accuracy in these downstream tasks, and to further evaluate pre-trained LMs versus fine-tuned LMs, we fine-tuned DistillBERT-base-uncased with the ImpPress dataset (Jeretic et al., 2020). It consists of >25k semi-automatically generated sentence pairs illustrating well-studied commonsense pragmatic inference types. 14100 tagged utterance pairs were used in the training of Presp, and 1410 tagged pairs for testing. Here is the input representation: sentence 1 *Victoria's mall that has hurt Sam might upset Helen.*; sentence 2 *Victoria doesn't have exactly one mall that has hurt Sam.*; Label *contradiction*. As to SI, 6000 tagged utterance pairs were used for training and 600 for testing. Here is the input representation: sentence 1 *The teacher resembles some sketches.*; sentence 2 *The teacher doesn't resemble all sketches.*; Label *entailment*.

We fine-tuned DistillBERT-base-uncased on an Apple M1 CPU for 3 epochs. We used a batch size 64 of and optimized using Adam (Kingma and Ba, 2014) with betas=(0.9,0.999), with a learning rate

---

[2]This would seem to raise questions about the strength of the conclusions being drawn (c.f. section 5) - it seems that LMs merely leverage co-occurrence frequency; on the other hand, it also appears that LMs' trend aligns with joint frequency - LMs does not fail the sanity check because frequency/prevalence heavily influences humans' commonsense reasoning too.

of 2*e*-05.

## 5  Evaluations and discussion

Error bar in Fig.1 shows DistillBERT does not seem to have difficulty detecting Presp, and fine-tuning slightly decreases its performance. This is likely due to the fact that Singh et al. (2016) data is not formatted the same as the ImpPress training data. Fine-tuning might have misled DistillBERT. Regarding SI, fine-tuning significantly increases LMs' performance, indicating that the ImpPress dataset is a good candidate for improving LMs' sensitivity to commonsense SIs. Error bar in Fig.2 indicates that GPT-3 is slightly better in detecting SI than in Presp, but overall GPT-3 is not good at the psycholinguistic task. This maybe because GPT-3 has a different architecture. LMs performance aligns with n-gram baseline in that overall the SI dataset is less challenging than the Presp: 70% of SI dataset shows the favorable co-occurrence direction: the pair tagged as 'good' also shows higher nouns co-occurrence rate than the 'bad' pair does. The Presp dataset is less helpful (50%).

It's worth noting that it's not clear if we can make a *direct* comparison between human decisions and LMs' rates, especially for the SI cases. Nieuwland et al. (2010) suggests that for humans, the informative and pragmatically good statements elicited larger N400 ERPs than underinformative and pragmatically bad statements. However, this does not directly transfer to the accuracy mean metric we used for LMs. All Fig.2 showed is that GPT-3's performance is roughly at chance, with respect to accuracy mean. For future studies, we plan to conduct parallel human studies to collect baseline human decision rates.

Regarding LMs evaluation analysis, our study shows that in order to probe commonsense knowledge from LMs, understand their reasoning mechanisms, and identify their limitations for AI applications due to the lack of commonsense knowledge, we need to carefully consider how to prompt the

pre-trained LMs. For masked LMs such as DistillBERT, our results suggest that an appropriate method to examine how 'human-like' LMs are is to mask the same token as psycholinguists do in their behavioral/neural experiments with humans, and keep the same contextual information, so that the experiment setting is as close to human experiments as possible. As to unidirectional LMs like GPT-3, they read in sentence using almost the same fundamental mechanisms as humans do, we thus took sentence to be a unit to derive logprob. How much GPT-3 like the sentence is directly reflected in its sentence logprob. It's crucial to use different metrics for BERT and GPT-3 to avoid the pitfall of comparing the two with the same metrics, as they are trained very differently, and a perplexity comparison would be inconclusive.

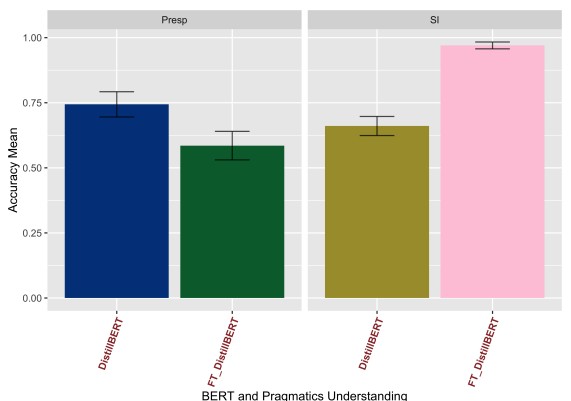

Figure 1: Evaluate BERT with human data. DistillBERT is used for critical word prediction. FT: fine-tuned.

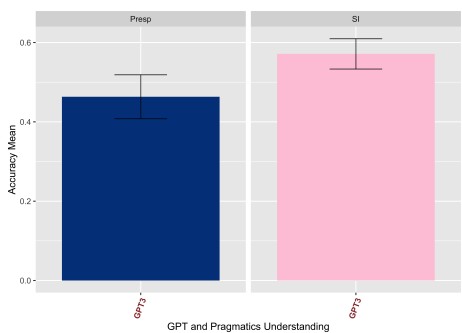

Figure 2: Evaluate GPT-3 with human data. GPT-3 is used for sequential word prediction.

Our study has some limitations. Although we mention multiple times that these pragmatics often exist in conversations, the actual datasets we used are not conversational. For future work, we hope to see how LMs perform in a conversation scenario in terms of commonsense pragmatics. This could give us a better grasp of LMs' competence at the conversational level of language understanding. For the current work, our motivation of using non-conversational human data for conversational implicature is that LMs are not trained the same way through many *dialogues*, but rather with text found on the web. Additionally, we acknowledge that there were some glitches in DistillBERT's SI evaluation setting. BERT is considered succeed as long as the critical word is in its top$K$. By not penalizing that *some* can be above *all* in the case where both would be in the top$K$ choices, we accept LM's choice as "correct" white it isn't. It's also not very surprising that *all* doesn't show up as much as other options in BERT's top$K$ choices for scenarios that *all* is the commonsense intent, given that LM might generate adjectives but not quantifiers to modify the following noun. It's likely that this has nothing to do with the implication, nevertheless they still make sense considering that the LM's learning algorithm uses masked loss. For future research, we hope to get more valid conclusions through directly comparing whether *all* is relatively more likely than *some*.

Humans show no difficulty in using commonsense knowledge to reason about daily conversations. By contrast, the extent to which LMs are sensitive to commonsense reasoning has remained an elusive research question in AI research for decades. Here, we provide an approach for commonsense reasoning tasks: incorporating online and offline psycholinguistic datasets into LMs evaluation. Using well-controlled task design and high resolution neurophysiology equipment, psycholinguistics studies all kinds of implicit meanings in natural language. To examine how 'human-like' LMs can be, human data is the key. These methods can improve the interpretability and explainability of neural models for reasoning about implied yet commonsense message.

To sum up, our paper aims to evaluate DistillBERT and GPT-3's ability to make human-like pragmatic inferences, such as SI and Presp, through human behavioral and neural data. Findings show psycholinguistic datasets can help get a good grasp of LMs' accuracy in detecting commonsense reasoning. Our study adopted a theory-supported lens for investigating the often vaguely-defined "commonsense", and illustrated how to establish connection between commonsense reasoning in NLP and pragmatic semantics.

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
