# OpenReview forum: "Psycholinguistic Diagnosis of Language Models’ Commonsense Reasoning"
_aclweb.org/ACL/2022/Workshop/CSRR — ACL 2022 Workshop CSRR_

### Official Review · Reviewer_KJoU · 2022-03-23
**Contradicting conclusions & some issues with experiment setting**

**Rating:** 4
**Confidence:** 5

**Review:**

This short paper aims to evaluate DistillBERT and GPT-3’s ability to make human-like pragmatic inferences, such as Scalar Implicature (SI) and Presupposition (Presp), through human behavioral and neural data.

In Section 5, the authors claim that:

* DistillBERT has difficulty detecting Presp (a bit far from human rate), and its performance suffers further when fine-tuned on another Presp + Implicature dataset (ImpPress dataset [1]).

* DistillBERT has difficulty detecting SI, but its performance improves to human-level when fine-tuned on the ImpPress dataset [1].

* GPT-3 does not do well in either psycholinguistic task and fine-tuning helps.

### Strengths

* **Online human data motivation:** The paper’s motivation of using human data for conversational implicature is well-grounded. LMs are not trained the same way through many dialogues, but rather with text found on the web.

* **Good sanity check on noun co-occurrence:** The authors check whether the test datasets contain enough noun co-occurrence patterns that could make the LM find a likelihood pattern rather than reason to conclude what sentence is more plausible.

* **Avoids the pitfall of comparing BERT and GPT-3 with the same metrics:** This is a positive thing as they are trained very differently, and a perplexity comparison would be inconclusive.

### Weaknesses

* **Important definitions/examples are not very clear:**
    - If the reader doesn’t know what SI or Presp is, they are left with definitions and examples that aren’t very clear (ex: [line 50] “by using the determiner ‘the’ most people typically presuppose the existence of such a thing in the context.”)
    - Similarly, in [line 142], it seems as if the uniqueness of the waiter affects the context, whereas it’s the relevancy of the job to the place. If one said “a waiter,” the place could be a random location irrelevant to the person who is still unique but happens to hold that job.

* **DistillBERT’s SI implicature evaluation setting doesn’t help answer the question:**
    - By not penalizing that “some” can be above “all,” in the case where both would be in the top 5 choices, we accept the model’s choice as correct while it isn't.
    - It’s not surprising that “all” doesn’t show up as much as other options in the top 5 choices, as the model may put many adjectives like “Dirty,” “Tall,” and so forth. These have nothing to do with the implication, yet they still make sense given that the LM’s learning algorithm uses masked loss. Instead, comparing whether “all” is relatively more likely than “some” is more useful and would lead to more valid conclusions.

* **GPT-3’s success criteria calculation is not clear:** [Line 116-122] is a little confusing as to which mean we are calculating. It seems to be that the authors calculate the sequence log probability divided by the sentence length for several sentences and then calculate the mean of those.

* **Contradictions in conclusions:** While the results in Section 5 state that GPT-3 doesn’t do very well with psycholinguistic tasks [line 278], the authors say that LMs understand “the implied intent shared among most people” in the abstract [line 019-022]. This is either because I misunderstood something or because even human decisions are as low as the LMs’ rates. Including the human plausibility decision rates in the plots can make it easier to compare with LMs, although the authors may have stated these numbers at [lines 152-154].

### Style - points that didn’t affect the decision of acceptance/rejection but could be useful to the authors

* This paper may be a single-author paper. It’s still preferred if it’s written with “we” statements instead of “I” statements.

* There are many places where previously cited references aren’t referred to with LaTeX again when mentioned, and are instead typed out, which leads to typos (ex: [line 155]). In the future, it may lead to not being able to discern between different articles.

### References

[1] [Are Natural Language Inference Models IMPPRESsive? Learning IMPlicature and PRESupposition](https://aclanthology.org/2020.acl-main.768) (Jeretic et al., ACL 2020)

---

### Official Review · Reviewer_SiFm · 2022-03-23
**Interesting direction with analysis**

**Rating:** 6
**Confidence:** 3

**Review:**

This paper investigates whether transformer-based neural language models (LM) understand commonsense pragmatics and focus on presupposition and scalar implicature.

Strength:
- I really like the connection between commonsense reasoning in NLP and pragmatic semantics in this work. It provides a theory-supported lens for investigating the often vaguely-defined "common sense".
- The datasets chosen to probe LMs are well-chosen and the paper provides sufficient details which I appreciate as a reviewer.
- The experiments are direct and easy-to-follow.

Places to improve:
- The authors mention multiple times that these pragmatics often exist in conversations, but the actual dataset used to probe models is not conversational. I would be really interested to see how models perform in a conversation scenario in terms of commonsense pragmatics.
- I would be very curious to learn more about the authors' explanations on some of the experimental results and further analysis (if page limit allows)

---

### Official Review · Reviewer_CQLG · 2022-03-25
**Analysis of LM understanding of specific presuppositions and implicatures**

**Rating:** 5
**Confidence:** 4

**Review:**

Strengths
- Probing LM understanding of pragmatic meanings is an interesting area and the paper presents a new approach to doing this
- the results as presented are promising, using data from cognitive science

Weaknesses
- there are many missing details that make parts of the paper difficult to understand (see questions below)
- it is difficult to fully understand the results, due to lack of clarity around the experiment details (especially the metrics) and what is being computed. There is also an issue of whether the results are affected by co-occurrence, which is mentioned but not addressed.
- There is no analysis of the types of errors the model is making, or even what the model outputs look like on examples. These are needed to fully understand the results and approach


Questions
L049-050: What is this example with "the" saying? Is it just an example of a presupposition? If so, this should be made clear
L052-053: Similarly to above, its not clear what the "some" sentence means.
L081: what is N400? why is this relevant?
L106: what is "contrastive distribution"?
L117: what is "percentage mean"?
L130: the experiments and data from Singh et al. are mentioned without any description. What is this study? What was it doing? We need this information to understand the end of the paragraph around L147.
L200: why is this the measure of pragmatic skill?
L212: what is the data from Nieuwland et al?
L238: this would seem to raise questions about the strength of the conclusions being drawn. Why is this not the case?


Typos/suggestions:
L024: "I" --> "we" (and throughout the paper)
L052-053: "most people generally implies" is not grammatical
L074: "survey" --> "surveys"
L083-084: this sentence is not grammatical
L097: "meaning" --> "meanings"
L129: "card" --> "cards"
L166: "shool" --> "school"
L184 and L186: "succeed" --> "to have succeeded"

---

### Decision · Program_Chairs · 2022-03-28

Accept